# Neural Multi-Objective Combinatorial Optimization via Graph-Image Multimodal Fusion

**Jinbiao Chen**[1], **Jiahai Wang**[1,2,3,*], **Zhiguang Cao**[4], **Yaoxin Wu**[5],

[1]School of Computer Science and Engineering, Sun Yat-sen University, P.R. China
[2]Key Laboratory of Machine Intelligence and Advanced Computing, Ministry of Education,
Sun Yat-sen University, P.R. China
[3]Guangdong Key Laboratory of Big Data Analysis and Processing, Guangzhou, P.R. China
[4]School of Computing and Information Systems, Singapore Management University, Singapore
[5]Department of Industrial Engineering & Innovation Sciences, Eindhoven University of Technology
chenjb69@mail2.sysu.edu.cn, wangjiah@mail.sysu.edu.cn
zgcao@smu.edu.sg, y.wu2@tue.nl

## Abstract

Existing neural multi-objective combinatorial optimization (MOCO) methods still exhibit an optimality gap since they fail to fully exploit the intrinsic features of problem instances. A significant factor contributing to this shortfall is their reliance solely on graph-modal information. To overcome this, we propose a novel graph-image multimodal fusion (GIMF) framework that enhances neural MOCO methods by integrating graph and image information of the problem instances. Our GIMF framework comprises three key components: (1) a constructed instance image to better represent the spatial structure of the problem instance, (2) a problem-size adaptive resolution strategy during the image construction process to improve the cross-size generalization of the model, and (3) a multimodal fusion mechanism with modality-specific bottlenecks to efficiently couple graph and image information. We demonstrate the versatility of our GIMF by implementing it with two state-of-the-art neural MOCO backbones. Experimental results on classic MOCO problems show that our GIMF significantly outperforms state-of-the-art neural MOCO methods and exhibits superior generalization capability.

## 1 Introduction

Multi-objective combinatorial optimization (MOCO) involves optimizing multiple conflicting objectives within a discrete solution space, resulting in trade-offs between different criteria for informative decision making. It holds significant importance due to its wide range of real-world applications in logistics, scheduling, resource allocation and so on (Ehrgott & Gandibleux, 2000; Liu et al., 2020; Türkyılmaz et al., 2020). MOCO aims to identify a *Pareto-optimal* set of solutions, where any improvement in one objective necessitates a compromise in another. Since an MOCO problem is more complex than its single-objective counterpart, exact methods are generally impractical for MOCO, due to the NP-hard complexity of typical CO problems. As an alternative, heuristic methods have been developed to efficiently search for approximate Pareto-optimal solutions. Nevertheless, the conventional heuristics heavily hinge on problem-specific expertise and instance-specific tuning work for achieving desirable performance, posing challenges in automatic algorithm development.

With the rapid progress of *neural CO* methods for single-objective CO problems (Kool et al., 2019; Kwon et al., 2020; Bi et al., 2022; Chen et al., 2022; Zhang et al., 2022; Grinsztajn et al., 2023; Chalumeau et al., 2023; Drakulic et al., 2023; Son et al., 2023; Fang et al., 2024; Xiao et al., 2024a;b;c; Goh et al., 2024; Liu et al., 2024; Zhou et al., 2024; Wang et al., 2024; Kong et al., 2024), recent years have witnessed a surge in the development of *neural MOCO* methods. Differing from conventional heuristics, the neural MOCO employs deep neural models to autonomously learn constructive policies from problem instances in a data-driven manner. It allows for the automatic search of promising end-to-end solutions, bypassing the labor-intensive algorithm development in

---

*Corresponding Author.

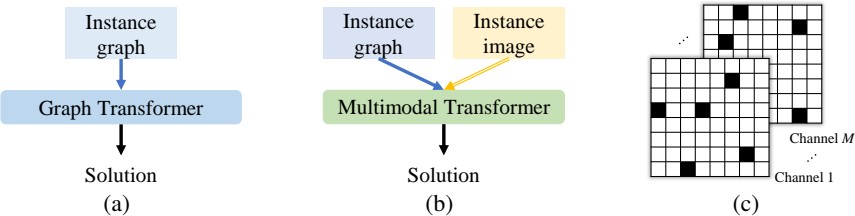

Figure 1: Typical neural MOCO methods (a) vs. the proposed GIMF framework (b). An example of an instance image (c) for an $M$-objective 5-node traveling salesman problem is illustrated, with blank and black pixels filled with 1 and 0, respectively.

conventional heuristics. However, current neural MOCO methods still exhibit an optimality gap, as they fail to fully exploit intrinsic features of problem instances. They commonly represent a problem instance as a graph that lacks the informed representations inherent in other modalities such as images, thus hindering neural models from achieving a comprehensive encoding.

In this work, we construct images of MOCO problems with the aim to offer the complementary instance information from different modalities, i.e., graphs and images. To advance neural models, we develop a *graph-image multimodal fusion* (GIMF) framework to synergize the multimodal representation learning for enhancing the MOCO performance. The difference between our framework and current neural MOCO methods, and the images used in this paper are illustrated in Figure 1.

Our contributions are summarized as follows. (1) We propose a transformation from an MOCO instance to construct an *instance image*, which with the graph representation, provides complementary multimodal information to facilitate comprehensive representation learning in neural models. (2) During the image construction, we present a *problem-size adaptive resolution* (PSAR) strategy that effectively enhances the model's generalization capacity for out-of-distribution problem sizes. (3) We design a multimodal fusion mechanism with *modality-specific bottlenecks* (MSB) to favorably synergize the multimodal information from the graph and image. (4) We demonstrate the versatility of our GIMF framework by deploying it with two state-of-the-art neural MOCO methods. Experimental results on classic MOCO problems show the significant superiority of our GIMF. The advantageous effects of the multimodal fusion and generalizability are corroborated as well.

## 2 RELATED WORKS

**Conventional MOCO methods.** Conventional MOCO methods are categorized into exact and heuristic approaches. Exact methods exhaustively identify Pareto-optimal solutions, resulting in prohibitive exponential computational complexity (Ehrgott et al., 2016; Figueira et al., 2017). In contrast, heuristic methods, particularly multi-objective evolutionary algorithms (MOEAs) (Deb et al., 2002; Zhang & Li, 2007; Deb & Jain, 2013; Deng et al., 2022; Qi et al., 2014; Yuan et al., 2016), are competent to efficiently search for near-optimal solutions. Furthermore, the problem-specific local search algorithms can be integrated into general MOEAs (Jaszkiewicz, 2002; Shi et al., 2020; 2024) to enhance the performance. Despite considerable domain knowledge invested in algorithm design, these conventional heuristics still require intensive intrinsic searches from scratch when solving each instance, underscoring a significant limitation in their effectiveness. For more detailed survey on heuristics for MOCO, please refer to Verma et al. (2021); Liu et al. (2020).

**Neural MOCO methods.** Most neural MOCO methods decompose the MOCO problem into a series of scalarized subproblems, addressing each through a single-objective neural CO method, such as the notable policy optimization with multiple optima (POMO) (Kwon et al., 2020). Based on the number of trained models, neural MOCO methods can be grouped into three categories: multi-model, single-model, and unified-model methods. Multi-model methods train a set of neural models via transfer learning (Li et al., 2021; Zhang et al., 2021) or meta-learning (Zhang et al., 2023; Chen et al., 2023a), with each model specialized for each specific subproblem. In contrast, single-model methods (Ye et al., 2022; Wang et al., 2024) employ a single model for all subproblems, typically represented by preference-conditioned multi-objective combinatorial optimization (PMOCO) (Lin et al., 2022), but they still require substantial efforts in the separate training for different problem

sizes. The most recent unified-model method, the conditional neural heuristic (CNH) (Fan et al., 2024), trains only a generic model that can generalize across various sizes, achieving the state-of-the-art performance in neural MOCO. Distinct from the mainstream decomposition-based neural methods, some orthogonal research attempts to enhance diversity (Chen et al., 2023b) or develop learning-based iterative methods (Wu et al., 2022; Su et al., 2024; Ye et al., 2025), which significantly increase computational overhead for limited improvements. This paper focuses on improving the optimality of subproblems in decomposition-based neural methods by employing the graph-image multimodal fusion.

## 3  PRELIMINARY

An MOCO problem can be defined as $\min_{\boldsymbol{\pi}\in\Omega}\boldsymbol{f}(\boldsymbol{\pi})=(f_1(\boldsymbol{\pi}),f_2(\boldsymbol{\pi}),\ldots,f_M(\boldsymbol{\pi}))$, where $\boldsymbol{f}$ is the objective vector with $M$ objective functions, $\boldsymbol{\pi}$ is the decision variable, and $\Omega$ is a discrete feasible solution space. The Pareto properties of solutions for an MOCO problem are provided as below.

**Definition 1 (Pareto dominance).** A solution $\boldsymbol{\pi}^1$ dominates another $\boldsymbol{\pi}^2$ (denoted as $\boldsymbol{\pi}^1\prec\boldsymbol{\pi}^2$) if $f_i(\boldsymbol{\pi}^1)\leq f_i(\boldsymbol{\pi}^2),\forall i\in\{1,\ldots,M\}$ and $f_j(\boldsymbol{\pi}^1)<f_j(\boldsymbol{\pi}^2),\exists j\in\{1,\ldots,M\}$.

**Definition 2 (Pareto optimality).** A solution $\boldsymbol{\pi}^*$ is Pareto optimal if it is not dominated by any other solution $\boldsymbol{\pi}$. The *Pareto set* refers to all Pareto optimal solutions, i.e., $\mathcal{P}=\{\boldsymbol{\pi}^*\in\Omega\mid\nexists\boldsymbol{\pi}'\in\Omega:\boldsymbol{\pi}'\prec\boldsymbol{\pi}^*\}$. Its image in the objective space is known as the *Pareto front*, i.e., $\mathcal{F}=\{\boldsymbol{f}(\boldsymbol{\pi})\in\mathcal{R}^M\mid\boldsymbol{\pi}\in\mathcal{P}\}$.

### 3.1  DECOMPOSITION-BASED NEURAL MOCO

The decomposition is a widely used technique for MOCO problems. An MOCO problem can be decomposed into $N$ subproblems, each of which is a scalarized CO problem associated with a weight vector $\boldsymbol{\lambda}\in\mathcal{R}^M$ satisfying $\lambda_i\geq 0,\forall i\in\{1,\ldots,M\}$ and $\sum_{i=1}^M\lambda_i=1$. The scalarized objective $g(\boldsymbol{\pi}|\boldsymbol{\lambda})$ for a subproblem can be derived by different scalarization functions, e.g., the most straight-forward weighted sum (WS) represents the objective $\min_{\boldsymbol{\pi}\in\Omega}g_{\mathrm{ws}}(\boldsymbol{\pi}|\boldsymbol{\lambda})=\sum_{i=1}^M\lambda_i f_i(\boldsymbol{\pi})$. After decomposition with $N$ weight vectors, the $N$ derived subproblems are solved by neural CO methods.

**Neural CO methods for subproblems.** Treating the solution $\boldsymbol{\pi}$ as a sequence $\boldsymbol{\pi}=\{\pi_1,\ldots,\pi_T\}$ of length $T$, the solution construction process for a scalarized subproblem is a Markov decision process, with following definitions: (1) The *state* at step $t\in\{1,\ldots,T\}$ consists of the weight vector $\boldsymbol{\lambda}$, the current partial solution $\boldsymbol{\pi}_{1:t-1}$, and the instance $\mathcal{G}$. (2) The *action* is selecting a node $\pi_t$ to add to $\boldsymbol{\pi}_{1:t-1}$. (3) The *state transition* is represented as $\boldsymbol{\pi}_{1:t}=\{\boldsymbol{\pi}_{1:t-1},\pi_t\}$. (4) The *reward* is defined as the negative of the scalarized objective, i.e., $R=-g(\boldsymbol{\pi}|\mathcal{G},\lambda)$. (5) The stochastic *policy* is parameterized by a neural model $\boldsymbol{\theta}$ and used to sequentially construct the solution, with the process denoted by $P(\boldsymbol{\pi}|\boldsymbol{\lambda},\mathcal{G})=\prod_{t=1}^T P_{\boldsymbol{\theta}}(\pi_t|\boldsymbol{\pi}_{1:t-1},\boldsymbol{\lambda},\mathcal{G})$. The policy network is typically trained by the REINFORCE algorithm (Williams, 1992), with the gradient $\nabla\mathcal{L}(\boldsymbol{\theta})=\frac{1}{B}\sum_{i=1}^B[(g(\boldsymbol{\pi}_i|\boldsymbol{\lambda},\mathcal{G}_i)-b)\nabla_{\boldsymbol{\theta}}\log P(\boldsymbol{\pi}_i|\boldsymbol{\lambda},\mathcal{G}_i)]$, where $B$ is the batch size, and $b$ is a baseline that is often derived from the average of multiple optima as done in POMO (Kwon et al., 2020), for reducing the variance.

**Graph Transformer.** An MOCO instance can be defined over a *graph* $\mathcal{G}=\{\mathcal{V},\mathcal{E}\}$, where $\mathcal{V}=\{v_1,\ldots,v_n\}$ denotes the node set and $\mathcal{E}=\{e(v_i,v_j)|v_i,v_j\in\mathcal{V},i\neq j\}$ denotes the edge set. Given an *instance graph* with $n$ nodes featured by $z$-dimensional vectors $\boldsymbol{u}_1,\ldots,\boldsymbol{u}_n\in\mathcal{R}^z$ (see Appendix A), the majority of existing neural MOCO methods all adopt a vanilla *graph Transformer* (Kool et al., 2019) (details in Appendix B) to handle the graph input. Generally, $L$ self-attention layers in encoder evolve node embeddings to $\boldsymbol{h}_1^{(L)},\ldots,\boldsymbol{h}_n^{(L)}\in\mathcal{R}^d$ ($d=128$). Then, the decoder uses the attention to autoregressively infer the probability of node selection with $T$ steps.

## 4  METHODOLOGY

Our graph-image multimodal fusion (GIMF) framework integrates complementary information from both graph and image modalities. The two key challenges are the construction of informative images from MOCO instances and the effective fusion of graph and image data. Correspondingly, we propose an image construction approach with a problem-size adaptive resolution (PSAR) strategy, and design a multimodal fusion mechanism with modality-specific bottlenecks (MSB).

## 4.1 IMAGE CONSTRUCTION

Before stepping into the image construction approach, we first introduce the definition of the image.

**Definition 3 (Image).** An *image* is defined as a discrete function $\mathcal{I} : \mathcal{D} \rightarrow \mathcal{C}^K$, where $\mathcal{D} = \{(x, y) | x, y \in \mathcal{Z}, 1 \leq x \leq W, 1 \leq y \leq H\}$ represents the set of *pixels*, with $(x, y)$ denoting a pixel, $W$ denoting the image width, and $H$ denoting the image height. The parameter $K$ refers to the number of *channels*, and $\mathcal{C}$ is the set of possible *pixel values* for each channel. The image size (a.k.a., *resolution*) is defined as $W \times H$. The most ubiquitous example is the 3-channel RGB image, where $\mathcal{C}^3 = [0, 255]^3$ represents the intensities in the red, green, and blue channels.

**Instance image for MOCO.** Given an MOCO instance, we construct its image by specifying node features as *pixels* and *pixel values* separately, referred to as the *instance image*. Our image construction is generally applicable to different MOCO problems. In this paper, we consider classic MOCO problems that are extensively studied in the neural MOCO literature (Lin et al., 2022; Chen et al., 2023a;b; Fan et al., 2024), including the bi/tri-objective traveling salesman problem (Bi/Tri-TSP) (Lust & Teghem, 2010), bi-objective capacitated vehicle routing problem (Bi-CVRP) (Zajac & Huber, 2021), and bi-objective knapsack problem (Bi-KP) (Ishibuchi et al., 2015) (Please see Appendix A for detailed problem statements). Their instance images are described below.

For the $M$-objective TSP, each node $i$, $\forall i \in \{1, \ldots, n\}$, is featured by $\boldsymbol{u}_i \in \mathcal{R}^{2M}$ representing $M$ groups of coordinates. Each pair $(v_{i,2j-1}, v_{i,2j})$, $\forall j \in \{1, \ldots, M\}$, corresponds to the $j$-th group of coordinates. An $M$-channel instance image $\mathcal{I}^{M-\text{TSP}}$ of the $M$-objective TSP, can be constructed by treating each coordinate as a pixel, and defined as follows,

$$\mathcal{I}_j^{M-\text{TSP}}(x, y) = \left\{ \begin{array}{ll} 0, & \text{if } (x, y) = (v'_{i,2j-1}, v'_{i,2j}) \\ 1, & \text{otherwise} \end{array} \right. , \tag{1}$$

where $(v'_{i,2j-1}, v'_{i,2j}) \in [1, W] \times [1, H]$ is a coordinate derived by normalizing the original $(v_{i,2j-1}, v_{i,2j}) \in [0, 1]^2$, that is, $(v'_{i,2j-1}, v'_{i,2j}) = (\lfloor W v_{i,2j-1} \rfloor + 1, \lfloor H v_{i,2j} \rfloor + 1)$. An example of $\mathcal{I}^{M-\text{TSP}}$ for a 5-node $M$-objective TSP instance is illustrated in Figure 1(c). For Bi-CVRP, each node feature $\boldsymbol{u}_i \in \mathcal{R}^3$ represents a coordinate $(v_1, v_2) \in [0, 1]^2$ and a demand $v_3 \in [0, 1)$. The single-channel instance image $\mathcal{I}^{\text{Bi}-\text{CVRP}}$ can be constructed by taking the normalized coordinate as a pixel and the demand as the pixel value. Formally, the image can be defined by $\mathcal{I}^{\text{Bi}-\text{CVRP}}(x, y) = v_3$, if $(x, y) = (v'_1, v'_2)$. For Bi-KP, the node feature $\boldsymbol{u}_i \in \mathcal{R}^3$ represents two values $(v_1, v_2) \in [0, 1]^2$ and a weight $v_3 \in (0, 1)$ of an item. Similarly, the single-channel instance image $\mathcal{I}^{\text{Bi}-\text{KP}}$ is constructed by taking two normalized values as a pixel and the weight as the pixel value, resulting in $\mathcal{I}^{\text{Bi}-\text{KP}}(x, y) = v_3$, if $(x, y) = (v'_1, v'_2)$.

## 4.2 PROBLEM-SIZE ADAPTIVE RESOLUTION

The constructed images are processed by a *vision Transformer* (Dosovitskiy et al., 2021), which is a mainstream neural model to handle image data. Concretely, a $K$-channel image with the resolution $W \times H$ is divided into non-overlapping patches with a uniform size $w \times h$. Each patch is flattened into a vector and linearly projected to a $d$-dimensional embedding. Positional encodings (e.g., the sinusoidal encoding) are added to the patch embeddings, enabling the model to be aware of the spatial relations between patches. Then, a standard Transformer encoder with $L$ self-attention layers evolves the embeddings to capture global features and contextual relationships within the image.

When our instance image adheres to the fixed resolution, a common setting in ordinary image-related tasks, it could pose challenges in the generalization across problem sizes. Specifically, we define the *node density* $\rho$ as the ratio of the non-one pixels in our instance image (Note: A low node density means that the number of non-one pixels approximately equals to the problem size $n$). Intuitively, the node density increases as the problem size $n$ grows with a fixed resolution, since more pixels are occupied by non-one values. The density at the patch level varies proportionally due to the fixed patch size. Consequently, the neural model is subject to the out-of-distribution generalization issue when the patch-level density significantly differs from those of the training data.

**Problem-size adaptive resolution (PSAR).** To address the above generalization issue, we propose a PSAR strategy to maintain a relatively stable density for both the image and patches. Generally, we can adaptively adjust the resolution by setting an approximately linear relationship between $W \times H$ and $n$ when constructing our instance image, so that the density is roughly uniform with varied

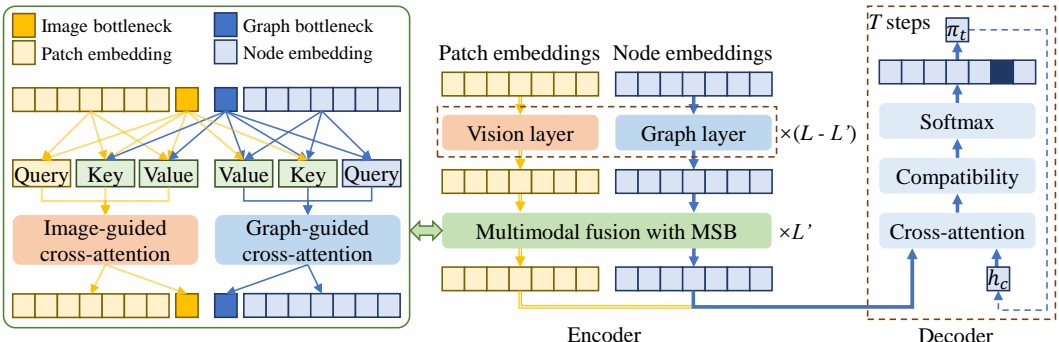

Figure 2: Our multimodal neural model based on an encoder-decoder (middle-right) architecture. The multimodal fusion layer with modality-specific bottlenecks (MSB) (left) leverages modality-specific bottlenecks to efficiently couple the graph and image information. The dashed line pointing from the selected node $\pi_t$ to the context embedding $\boldsymbol{h}_c$ indicates that $\boldsymbol{h}_c$ is determined by $\pi_t$.

problem sizes. In this paper, we set $w = h$ and $W = H = \lceil 10\sqrt{n}/w \rceil \times w$ to maintain the node density $\rho$ of approximately 0.01. By doing so, the number of patches (i.e., the length of the patch sequence) varies with the changing problem size. However, the vision Transformer benefits from the stable density among patches, and hence can gain better generalization performance.

**Scalable positional encoding.** With the fixed patch size, the PSAR results in varied lengths of the patch sequence. To indicate the positions of patches, we propose to learn scalable positional encodings. We utilize a multi-layer perceptron (MLP) with a $d$-dimensional hidden layer and ReLU activation to map the patch coordinates $(x^p, y^p) \in [0, 1]^2$ into position embeddings, which are then added to the patch embeddings. The MLP is applicable to varied problem sizes due to the fixed size of input (i.e., patches).

### 4.3 MULTIMODAL FUSION

The simple graph structures (e.g., fully-connected graphs for TSP, CVRP, and KP) may obscure spatial relationships among nodes. Our instance image offers a more informative view of the spatial structure, which maps a more explicit node distribution to an image. Meanwhile, the graph remains essential for fine-grained node-level encoding and decoding. To effectively integrate these two complementary modalities, we develop a multimodal fusion model with modality-specific bottlenecks.

Our multimodal fusion model, built in an encoder-decoder structure, is illustrated in Figure 2. Given the graph and instance image of an instance, the $L$-layer encoder processes them to generate high-dimensional node and patch embeddings. After that, the decoder computes the probabilities for node selection, which are used to sample nodes for constructing the solution.

**Single-modal layer.** In the encoder, the node and patch embeddings are first passed through $L - L'$ single-modal graph and vision layers, respectively. Specifically, the initial node embeddings $\boldsymbol{h}_1^{(0)}, \ldots, \boldsymbol{h}_n^{(0)} \in \mathcal{R}^d$ are obtained via linear projection with a trainable matrix $W^u$ and bias $\boldsymbol{b}^u$, which is formulated by $\boldsymbol{h}_i^{(0)} = W^u \boldsymbol{u}_i + \boldsymbol{b}^u, \forall i \in \{1, \ldots, n\}$. The initial patch embeddings $\boldsymbol{h}_1'^{(0)}, \ldots, \boldsymbol{h}_n'^{(0)} \in \mathcal{R}^d$ are derived from the flattened patch vectors $\boldsymbol{u}_1', \ldots, \boldsymbol{u}_{n'}' \in \mathcal{R}^{wh}$ via linear projection and position encoding, which is formulated by $\boldsymbol{h}_i'^{(0)} = W^{u'} \boldsymbol{u}_i' + \boldsymbol{b}^{u'} + \text{MLP}(x_i^p, y_i^p), \forall i \in \{1, \ldots, n'\}$, where $n'$ is the number of patches. Then, each single-modal graph and vision layer consists of a multi-head self-attention (MHSA) block with 8 heads and a feed-forward (FF) block. Each block is followed by the residual connection (He et al., 2016) and instance normalization (IN) (Ulyanov et al., 2016). The embeddings are updated at each layer $l, \forall l \in \{1, \ldots, L-L'\}$, as follows,

$$\hat{\boldsymbol{H}} = \text{IN}(\boldsymbol{H}^{(l-1)} + \text{MHSA}(\boldsymbol{H}^{(l-1)})), \quad \boldsymbol{H}^{(l)} = \text{IN}(\hat{\boldsymbol{H}} + \text{FF}(\hat{\boldsymbol{H}})),$$
$$\hat{\boldsymbol{H}}' = \text{IN}(\boldsymbol{H}'^{(l-1)} + \text{MHSA}(\boldsymbol{H}'^{(l-1)})), \quad \boldsymbol{H}'^{(l)} = \text{IN}(\hat{\boldsymbol{H}}' + \text{FF}(\hat{\boldsymbol{H}}')),$$

(2)

where $\boldsymbol{H}$ and $\boldsymbol{H}'$ denote the concatenation of the node embeddings $\{\boldsymbol{h}_1, \ldots, \boldsymbol{h}_n\}$ and the patch embeddings $\{\boldsymbol{h}_1', \ldots, \boldsymbol{h}_{n'}'\}$, respectively, which are iteratively evolved by single-modal layers.

**Multimodal layer.** After $L - L'$ single-modal layers, $L'$ multimodal fusion layers are used to efficiently synergize the graph and image information. We introduce a small set of bottleneck tokens to manage cross-modal interaction for condensing essential information and alleviating the computational burden of full attention calculations. Unlike the vanilla bottleneck approach (Nagrani et al., 2021), which uses shared bottlenecks across modalities, we propose modality-specific bottlenecks (MSB) to more precisely concentrate the information within each modality.

Specifically, we introduce $n_b$ ($n_b \ll n$) graph bottlenecks $\boldsymbol{B} = \{\boldsymbol{b}_1, \ldots, \boldsymbol{b}_{n_b}\}$ and $n'_b$ ($n'_b \ll n'$) image bottlenecks $\boldsymbol{B}' = \{\boldsymbol{b}'_1, \ldots, \boldsymbol{b}'_{n'_b}\}$ into the graph and vision layers, respectively. The initial graph and image bottlenecks $\boldsymbol{B}_i^{(L-L'+1)}, \boldsymbol{B}_i'^{(L-L'+1)} \in \mathcal{R}^d$ are random learnable parameters and updated through $L'$ multimodal fusion layers. Using the bottlenecks, the efficient fusion is achieved by two types of multi-head cross-attention (MHCA) with 8 heads. As depicted in Figure 2, the graph-guided and image-guided cross-attentions symmetrically fuse information between the two modalities. The multimodal cross-attentions at layer $l \in \{L - L' + 1, \ldots, L\}$, are given below,

$$
\begin{aligned}
\{\boldsymbol{H}^{(l)}, \boldsymbol{B}^{(l)}\} &= \mathrm{MHCA}(\{\boldsymbol{H}^{(l-1)}, \boldsymbol{B}^{(l-1)}\}, \{\boldsymbol{H}^{(l-1)}, \boldsymbol{B}^{(l-1)}, \boldsymbol{B}'^{(l-1)}\}), \\
\{\boldsymbol{H}'^{(l)}, \boldsymbol{B}'^{(l)}\} &= \mathrm{MHCA}(\{\boldsymbol{H}'^{(l-1)}, \boldsymbol{B}'^{(l-1)}\}, \{\boldsymbol{H}'^{(l-1)}, \boldsymbol{B}'^{(l-1)}, \boldsymbol{B}^{(l-1)}\}),
\end{aligned} \tag{3}
$$

where $\mathrm{MHCA}(\boldsymbol{X}, \boldsymbol{Y})$ denotes the multi-head cross-attention with $\boldsymbol{X}$ representing the *queries* and $\boldsymbol{Y}$ representing the *keys* and *values*. The modal-specific bottlenecks improve performance of multimodal fusion, while avoiding the computational burden by self-attention between all embeddings in modalities. We use the FF block, residual connection and IN similarly as in the single-modal layer.

**Probability for node selection.** In the decoder, we apply the node and patch embeddings processed by multimodal layers to autoregressively compute the probability for node selection. At each decoding step $t \in \{1, ..., T\}$, a *context* embedding $\boldsymbol{h}_c$, which depends on $\pi_{t-1}$ (see Appendix C), is used to compute a *glimpse* $\boldsymbol{q}_c$ via an MHCA block with 8 attention heads. This glimpse is then used to calculate the *compatibility* score $\boldsymbol{\alpha}$, as follows,

$$
\begin{aligned}
\boldsymbol{q}_c &= \mathrm{MHCA}(\boldsymbol{h}_c, \{\boldsymbol{H}^{(L)}, \boldsymbol{H}'^{(L)}\}), \\
\alpha_i &= \begin{cases} -\infty, & \text{if node } i \text{ is masked} \\ C \cdot \tanh\left(\frac{\boldsymbol{q}_c^T (W^K \boldsymbol{h}_i^{(L)})}{\sqrt{d}}\right), & \text{otherwise} \end{cases}
\end{aligned} \tag{4}
$$

where the result is clipped with $C = 10$ (Kool et al., 2019). Finally, the probabilities for selecting eligible nodes are normalized using softmax.

**Deployment onto neural MOCO methods.** Our GIMF is a generic framework that can be integrated with existing neural MOCO methods. We implement it with the state-of-the-art CNH to further enhance the performance in solving MOCO problems. Additionally, we implement GIMF with the well-known PMOCO, a type of method different from CNH, to demonstrate its flexibility. The resulting methods are referred to as GIMF-C and GIMF-P, respectively. GIMF-C employs a dual-attention model to associate the instance with the weight vector of the scalarized subproblem, while GIMF-P utilizes a hypernetwork to tackle the weight vector of the scalarized subproblem. The detailed architectures of GIMF-C and GIMF-P are given in Appendix D.

## 5 EXPERIMENTS

### 5.1 EXPERIMENTAL SETTINGS

**Problems.** We assess our GIMF framework on four well-established MOCO problems commonly examined in neural MOCO research, including Bi-TSP, Bi-CVRP, Bi-KP, and Tri-TSP. Detailed descriptions are provided in Appendix A. For the experiments, we use three standard instance sizes: $n = 20/50/100$ for Bi-TSP, Bi-CVRP, and Tri-TSP, and $n = 50/100/200$ for Bi-KP. While CNH and GIMF-C are trained across problem sizes $n \in \{20, 21, \cdots, 100\}$ (except $n \in \{50, 51, \cdots, 200\}$ for Bi-KP), other neural methods are trained separately for each problem size. Moreover, we evaluate the model's out-of-distribution generalization capability on larger instances of Bi-TSP150/200 and three widely used TSPLIB (Reinelt, 1991) benchmark instances, i.e., KroAB100/150/200.

**Hyperparameters.** Most hyperparameters for GIMF-P and GIMF-C are configured in line with the original PMOCO and CNH, respectively. For our model, $L = 6$, $L' = 3$, and $n_b = n'_b = 10$. The

patch dimensions are fixed at $w = h = 16$. The model undergoes training for 200 epochs, with each epoch processing 100,000 randomly selected instances and a batch size of $B = 64$. The Adam optimizer (Kingma & Ba, 2015) is used with a learning rate of $10^{-4}$ (except $10^{-5}$ for Bi-KP) and weight decay of $10^{-6}$. The $N$ weight vectors for the decomposition are generated according to Das & Dennis (1998), with $N = 101$ for $M = 2$ and $N = 105$ for $M = 3$.

**Baselines.** We compare our method against state-of-the-art neural MOCO methods, the widely used MOEAs, and strong problem-specific heuristics, as outlined below. (1) Neural MOCO methods: This includes the state-of-the-art **CNH** (Fan et al., 2024) and **PMOCO** (Lin et al., 2022). Additionally, we include multi-model methods such as **DRL-MOA** (Li et al., 2021), **MDRL** (Zhang et al., 2023), and **EMNH** (Chen et al., 2023a). In particular, DRL-MOA trains $N$ POMO models for the $N$ subproblems, starting with 200 epochs for the first model and using 5-epoch parameter transfer for each subsequent model. Both MDRL and EMNH fine-tune the $N$ POMO models from a shared pretrained meta-model with the same structure, following the training and fine-tuning configurations as described in Chen et al. (2023a). (2) Widely used MOEAs: Specifically, **MOEA/D** (Zhang & Li, 2007) and **NSGA-II** (Deb et al., 2002) are implemented with 4,000 iterations, representing decomposition-based and dominance-based MOEAs, respectively. **MOGLS** (Jaszkiewicz, 2002), also with 4,000 iterations and 100 local search steps per iteration, and **PPLS/D-C** (Shi et al., 2024), which runs for 200 iterations, are both tailored for MOCO with a 2-opt heuristic for TSP and CVRP and a greedy transformation heuristic (Ishibuchi et al., 2015) for KP. (3) Strong problem-specific heuristics: For the multi-objective TSP and KP, **WS-LKH** and **WS-DP** combine the weighted sum (WS) scalarization with the strong LKH (Tinós et al., 2018) and dynamic programming (DP) solvers to handle scalarized subproblems. All methods adopt WS scalarization for fair comparisons, and are executed on a machine equipped with an RTX 3090 GPU and an Intel Xeon 4216 CPU. Our code is publicly available[1].

**Metrics.** To assess the performance of the MOCO methods, we use the widely recognized hypervolume (HV) indicator (Audet et al., 2021), where a higher HV value indicates a superior solution set (see Appendix E for more details). We report the average HV, the gaps relative to GIMF-C-Aug, and the total computation time for 200 instances. Methods with "-Aug" incorporate instance augmentation (Lin et al., 2022) (details in Appendix F) to enhance performance. A Wilcoxon rank-sum test at a 1% significance level is conducted to assess statistical differences. The best result and those not significantly different from it are highlighted in **bold**, while the second-best result and those not significantly different from it are underlined. The names of our methods are also presented in **bold**.

## 5.2 EXPERIMENTAL RESULTS

**Comparison analysis.** The comparison results are presented in Tables 1 and 2. GIMF-C consistently outperforms CNH across all scenarios, establishing itself as the new state-of-the-art neural MOCO method. Similarly, GIMF-P shows significant improvement over PMOCO in every case, and it even surpasses PMOCO-Aug, which employs instance augmentation, on Bi-CVRP100, achieving a gap of 0.69% vs 2.70%. These results confirm the effectiveness of GIMF in synergizing the complementary strengths of graph and image information. Moreover, when enhanced with instance augmentation, GIMF's performance improves further. Compared with the cumbersome multi-model methods that require training or fine-tuning numerous models, GIMF-C maintains superiority in most cases, with only slight underperformance relative to EMNH on Bi-CVRP and Bi-KP. Notably, GIMF-C still manifests significant superiority over EMNH on Bi-TSP and Tri-TSP, such as a gap of 1.00% vs 3.09% on Tri-TSP100. Additionally, GIMF significantly reduces computational time compared with conventional methods, as demonstrated by GIMF-C-Aug, which requires just 21 minutes vs 6.0 hours for WS-LKH on Bi-TSP100, while delivering competitive results.

**Out-of-distribution size generalization analysis.** We evaluate the model's generalization capability on the out-of-distribution larger-size Bi-TSP150/200 instances and benchmark instances KroAB100/150/200. The results of the neural methods are obtained based on models trained on Bi-TSP100, except CNH and GIMF-C trained across sizes $n \in \{20, 21, \cdots, 100\}$, as shown in Table 3 and Figure 3. Compared with all neural baselines and well-known MOEAs, GIMF-C achieves the best generalization performance across all out-of-distribution cases. Similarly, GIMF-P consistently outperforms PMOCO in all scenarios. The visualized Pareto fronts of KroAB100/150/200 in Figure

---

[1] https://github.com/bill-cjb/GIMF

Table 1: Comparison results on Bi-TSP and Bi-CVRP both with 200 random instances.

| Method | Bi-TSP20 | | | Bi-TSP50 | | | Bi-TSP100 | | |
|---|---|---|---|---|---|---|---|---|---|
| | HV↑ | Gap↓ | Time↓ | HV↑ | Gap↓ | Time↓ | HV↑ | Gap↓ | Time↓ |
| WS-LKH | 0.6270 | 0.00% | 10m | **0.6415** | **-0.06%** | 1.8h | **0.7090** | **-0.35%** | 6.0h |
| MOEA/D | 0.6241 | 0.46% | 1.7h | 0.6316 | 1.48% | 1.8h | 0.6899 | 2.35% | 2.2h |
| NSGA-II | 0.6258 | 0.19% | 6.0h | 0.6120 | 4.54% | 6.1h | 0.6692 | 5.28% | 6.9h |
| MOGLS | **0.6279** | **-0.14%** | 1.6h | 0.6330 | 1.26% | 3.7h | 0.6854 | 2.99% | 11h |
| PPLS/D-C | 0.6256 | 0.22% | 26m | 0.6282 | 2.01% | 2.8h | 0.6844 | 3.13% | 11h |
| DRL-MOA | 0.6257 | 0.21% | 6s | 0.6360 | 0.80% | 9s | 0.6970 | 1.34% | 21s |
| MDRL | 0.6271 | -0.02% | 5s | 0.6364 | 0.73% | 9s | 0.6969 | 1.36% | 17s |
| EMNH | 0.6271 | -0.02% | 5s | 0.6364 | 0.73% | 9s | 0.6969 | 1.36% | 16s |
| PMOCO | 0.6259 | 0.18% | 6s | 0.6351 | 0.94% | 10s | 0.6957 | 1.53% | 19s |
| **GIMF-P** | 0.6266 | 0.06% | 7s | 0.6374 | 0.58% | 12s | 0.7006 | 0.84% | 24s |
| CNH | 0.6270 | 0.00% | 14s | 0.6387 | 0.37% | 17s | 0.7019 | 0.65% | 29s |
| **GIMF-C** | 0.6270 | 0.00% | 15s | 0.6397 | 0.22% | 19s | 0.7040 | 0.35% | 33s |
| MDRL-Aug | 0.6271 | -0.02% | 33s | 0.6408 | 0.05% | 1.7m | 0.7022 | 0.61% | 14m |
| EMNH-Aug | 0.6271 | -0.02% | 33s | 0.6408 | 0.05% | 1.7m | 0.7023 | 0.59% | 14m |
| PMOCO-Aug | 0.6270 | 0.00% | 1.1m | 0.6395 | 0.25% | 3.2m | 0.7016 | 0.69% | 15m |
| **GIMF-P-Aug** | 0.6271 | -0.02% | 1.5m | 0.6403 | 0.12% | 4.7m | 0.7043 | 0.31% | 20m |
| CNH-Aug | 0.6271 | -0.02% | 1.5m | 0.6410 | 0.02% | 4.1m | 0.7054 | 0.16% | 16m |
| **GIMF-C-Aug** | 0.6270 | 0.00% | 2.0m | 0.6411 | 0.00% | 5.5m | 0.7065 | 0.00% | 21m |

| Method | Bi-CVRP20 | | | Bi-CVRP50 | | | Bi-CVRP100 | | |
|---|---|---|---|---|---|---|---|---|---|
| | HV↑ | Gap↓ | Time↓ | HV↑ | Gap↓ | Time↓ | HV↑ | Gap↓ | Time↓ |
| MOEA/D | 0.4255 | 1.05% | 2.3h | 0.4000 | 2.49% | 2.9h | 0.3953 | 3.09% | 5.0h |
| NSGA-II | 0.4275 | 0.58% | 6.4h | 0.3896 | 5.02% | 8.8h | 0.3620 | 11.25% | 9.4h |
| MOGLS | 0.4278 | 0.51% | 9.0h | 0.3984 | 2.88% | 20h | 0.3875 | 5.00% | 72h |
| PPLS/D-C | 0.4287 | 0.30% | 1.6h | 0.4007 | 2.32% | 9.7h | 0.3946 | 3.26% | 38h |
| DRL-MOA | 0.4287 | 0.30% | 10s | 0.4076 | 0.63% | 12s | 0.4055 | 0.59% | 33s |
| MDRL | 0.4291 | 0.21% | 8s | 0.4082 | 0.49% | 13s | 0.4056 | 0.56% | 32s |
| EMNH | 0.4299 | 0.02% | 7s | 0.4098 | 0.10% | 13s | 0.4072 | 0.17% | 31s |
| PMOCO | 0.4267 | 0.77% | 7s | 0.4036 | 1.61% | 12s | 0.3913 | 4.07% | 27s |
| **GIMF-P** | 0.4287 | 0.30% | 7s | 0.4076 | 0.63% | 12s | 0.4051 | 0.69% | 29s |
| CNH | 0.4287 | 0.30% | 15s | 0.4087 | 0.37% | 17s | 0.4065 | 0.34% | 31s |
| **GIMF-C** | 0.4289 | 0.26% | 15s | 0.4089 | 0.32% | 18s | 0.4066 | 0.32% | 36s |
| MDRL-Aug | 0.4294 | 0.14% | 11s | 0.4092 | 0.24% | 36s | 0.4072 | 0.17% | 2.8m |
| EMNH-Aug | **0.4302** | **-0.05%** | 11s | **0.4106** | **-0.10%** | 35s | **0.4079** | **0.00%** | 2.8m |
| PMOCO-Aug | 0.4294 | 0.14% | 14s | 0.4080 | 0.54% | 36s | 0.3969 | 2.70% | 2.7m |
| **GIMF-P-Aug** | 0.4298 | 0.05% | 17s | 0.4098 | 0.10% | 48s | 0.4075 | 0.10% | 3.0m |
| CNH-Aug | 0.4299 | 0.02% | 22s | 0.4101 | 0.02% | 45s | 0.4077 | 0.05% | 2.5m |
| **GIMF-C-Aug** | 0.4300 | 0.00% | 25s | 0.4102 | 0.00% | 54s | **0.4079** | **0.00%** | 3.1m |

3 clearly show that the solutions identified by GIMF-C (or GIMF-P) are superior to those found by CNH (or PMOCO). Detailed results on these benchmark instances are provided in Appendix G.

**Effectiveness of the multimodal fusion mechanism with MSB.** To verify the effectiveness of our MSB, we compare GIMF with models utilizing other multimodal fusion mechanisms. The baselines include the vanilla self-attention (VSA) using a unified Transformer stream, the vanilla cross-attention (VCA) with two dedicated Transformer streams, a variant of MSB using fully learnable bottlenecks (FLB) in each layer, and the vanilla shared bottlenecks (SB) (Nagrani et al., 2021) with the same total number of bottlenecks as MSB. As shown in Figure 4, the results on in-distribution Bi-TSP100 demonstrate that all multimodal methods outperform CNH, although the differences among these methods are not significant. However, the results differ on the out-of-distribution Bi-TSP200, where GIMF performs the best, while VCA, the worst-performing model, even falls short of CNH. In conclusion, while incorporating image information improves in-distribution performance, our multimodal fusion mechanism with MSB is critical for superior out-of-distribution performance.

**Effectiveness of the PSAR strategy.** To assess the effectiveness of the PSAR strategy, we remove it from both GIMF and VSA, resulting in the variants GIMF w/o PSAR and VSA w/o PSAR. For

Table 2: Comparison results on Bi-KP and Tri-TSP both with 200 random instances.

| Method | Bi-KP50 | | | Bi-KP100 | | | Bi-KP200 | | |
|---|---|---|---|---|---|---|---|---|---|
| | HV↑ | Gap↓ | Time↓ | HV↑ | Gap↓ | Time↓ | HV↑ | Gap↓ | Time↓ |
| WS-DP | **0.3561** | **-0.03%** | 22m | 0.4532 | 0.00% | 2.0h | 0.3601 | 0.03% | 5.8h |
| MOEA/D | 0.3540 | 0.56% | 1.6h | 0.4508 | 0.53% | 1.7h | 0.3581 | 0.58% | 1.8h |
| NSGA-II | 0.3547 | 0.37% | 7.8h | 0.4520 | 0.26% | 8.0h | 0.3590 | 0.33% | 8.4h |
| MOGLS | 0.3540 | 0.56% | 5.8h | 0.4510 | 0.49% | 10h | 0.3582 | 0.56% | 18h |
| PPLS/D-C | 0.3528 | 0.90% | 18m | 0.4480 | 1.15% | 47m | 0.3541 | 1.69% | 1.5h |
| DRL-MOA | 0.3559 | 0.03% | 9s | 0.4531 | 0.02% | 18s | 0.3601 | 0.03% | 1.0m |
| MDRL | 0.3530 | 0.84% | 6s | 0.4532 | 0.00% | 21s | 0.3601 | 0.03% | 1.2m |
| EMNH | **0.3561** | **-0.03%** | 6s | **0.4535** | **-0.07%** | 21s | **0.3603** | **-0.03%** | 1.2m |
| PMOCO | 0.3552 | 0.22% | 9s | 0.4523 | 0.20% | 19s | 0.3595 | 0.19% | 1.3m |
| **GIMF-P** | 0.3560 | 0.00% | 9s | 0.4533 | -0.02% | 21s | 0.3602 | 0.00% | 1.3m |
| CNH | 0.3556 | 0.11% | 18s | 0.4527 | 0.11% | 27s | 0.3598 | 0.11% | 1.2m |
| **GIMF-C** | 0.3560 | 0.00% | 18s | 0.4532 | 0.00% | 29s | 0.3602 | 0.00% | 1.4m |

| Method | Tri-TSP20 | | | Tri-TSP50 | | | Tri-TSP100 | | |
|---|---|---|---|---|---|---|---|---|---|
| | HV↑ | Gap↓ | Time↓ | HV↑ | Gap↓ | Time↓ | HV↑ | Gap↓ | Time↓ |
| WS-LKH | **0.4712** | **-0.13%** | 12m | **0.4440** | **-0.54%** | 1.9h | **0.5076** | **-1.10%** | 6.6h |
| MOEA/D | 0.4702 | 0.08% | 1.9h | 0.4314 | 2.31% | 2.2h | 0.4511 | 10.16% | 2.4h |
| NSGA-II | 0.4238 | 9.94% | 7.1h | 0.2858 | 35.28% | 7.5h | 0.2824 | 43.76% | 9.0h |
| MOGLS | 0.4701 | 0.11% | 1.5h | 0.4211 | 4.64% | 4.1h | 0.4254 | 15.28% | 13h |
| PPLS/D-C | 0.4698 | 0.17% | 1.4h | 0.4174 | 5.48% | 3.9h | 0.4376 | 12.85% | 14h |
| DRL-MOA | 0.4699 | 0.15% | 6s | 0.4303 | 2.56% | 9s | 0.4806 | 4.28% | 19s |
| MDRL | 0.4699 | 0.15% | 5s | 0.4317 | 2.24% | 9s | 0.4852 | 3.37% | 16s |
| EMNH | 0.4699 | 0.15% | 5s | 0.4324 | 2.08% | 9s | 0.4866 | 3.09% | 16s |
| PMOCO | 0.4693 | 0.28% | 5s | 0.4315 | 2.29% | 8s | 0.4858 | 3.25% | 18s |
| **GIMF-P** | 0.4702 | 0.08% | 6s | 0.4354 | 1.40% | 10s | 0.4927 | 1.87% | 23s |
| CNH | 0.4698 | 0.17% | 10s | 0.4358 | 1.31% | 14s | 0.4931 | 1.79% | 26s |
| **GIMF-C** | 0.4702 | 0.08% | 12s | 0.4384 | 0.72% | 16s | 0.4971 | 1.00% | 29s |
| MDRL-Aug | **0.4712** | **-0.13%** | 2.6m | 0.4408 | 0.18% | 25m | 0.4958 | 1.25% | 1.7h |
| EMNH-Aug | **0.4712** | **-0.13%** | 2.6m | 0.4418 | -0.05% | 25m | 0.4973 | 0.96% | 1.7h |
| PMOCO-Aug | **0.4712** | **-0.13%** | 5.1m | 0.4409 | 0.16% | 28m | 0.4956 | 1.29% | 1.7h |
| **GIMF-P-Aug** | **0.4712** | **-0.13%** | 13m | 0.4415 | 0.02% | 42m | 0.5001 | 0.40% | 2.7h |
| CNH-Aug | 0.4704 | 0.04% | 8.0m | 0.4409 | 0.16% | 33m | 0.4996 | 0.50% | 2.1h |
| **GIMF-C-Aug** | 0.4706 | 0.00% | 16m | 0.4416 | 0.00% | 44m | 0.5021 | 0.00% | 2.9h |

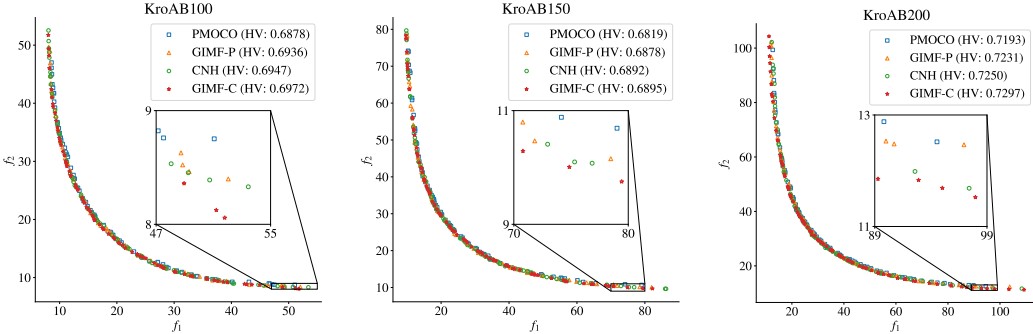

Figure 3: Pareto fronts of benchmark instances, KroAB100/150/200 (left/middle/right).

these variants, we set $W = H = \lceil 100/w \rceil \times w$, consistent with the size when $n = 100$. As shown in Figure 5, PSAR leads to a slight performance improvement on the in-distribution Bi-TSP100. More importantly, it significantly boosts the out-of-distribution performance for both GIMF and VSA. Notably, without PSAR, GIMF and VSA even fall below CNH's generalization capability. These results underscore the critical role of the PSAR strategy in enhancing out-of-distribution performance.

Table 3: Out-of-distribution generalization on larger-size problems with 200 random instances.

| Method | Bi-TSP150 | | | Bi-TSP200 | | |
|---|---|---|---|---|---|---|
| | HV↑ | Gap↓ | Time↓ | HV↑ | Gap↓ | Time↓ |
| WS-LKH | **0.7149** | **-1.65%** | 13h | **0.7490** | **-1.74%** | 22h |
| MOEA/D | 0.6809 | 3.18% | 2.4h | 0.7139 | 3.03% | 2.7h |
| NSGA-II | 0.6659 | 5.32% | 6.8h | 0.7045 | 4.31% | 6.9h |
| MOGLS | 0.6768 | 3.77% | 22h | 0.7114 | 3.37% | 38h |
| PPLS/D-C | 0.6784 | 3.54% | 21h | 0.7106 | 3.48% | 32h |
| DRL-MOA | 0.6901 | 1.88% | 45s | 0.7219 | 1.94% | 1.5m |
| MDRL | 0.6922 | 1.58% | 40s | 0.7251 | 1.51% | 1.4m |
| EMNH | 0.6930 | 1.46% | 40s | 0.7260 | 1.39% | 1.4m |
| PMOCO | 0.6910 | 1.75% | 50s | 0.7231 | 1.78% | 1.5m |
| **GIMF-P** | 0.6958 | 1.07% | 60s | 0.7267 | 1.29% | 2.1m |
| CNH | 0.6985 | 0.68% | 1.1m | 0.7292 | 0.95% | 1.9m |
| **GIMF-C** | 0.6993 | 0.57% | 1.2m | 0.7325 | 0.50% | 2.2m |
| MDRL-Aug | 0.6976 | 0.81% | 47m | 0.7299 | 0.86% | 1.6h |
| EMNH-Aug | 0.6983 | 0.71% | 47m | 0.7307 | 0.75% | 1.6h |
| PMOCO-Aug | 0.6967 | 0.94% | 47m | 0.7283 | 1.07% | 1.6h |
| **GIMF-P-Aug** | 0.7003 | 0.43% | 1.0h | 0.7311 | 0.69% | 2.1h |
| CNH-Aug | 0.7025 | 0.11% | 52m | 0.7343 | 0.26% | 1.7h |
| **GIMF-C-Aug** | 0.7033 | 0.00% | 1.1h | 0.7362 | 0.00% | 2.2h |

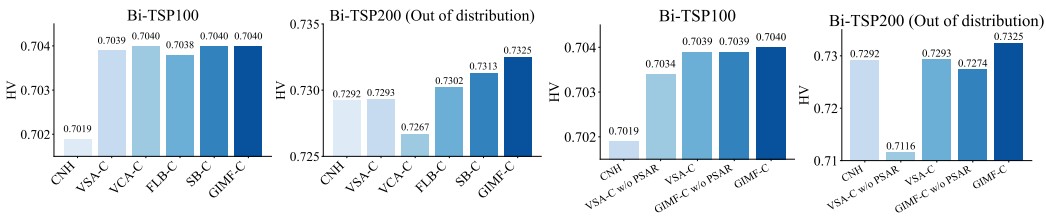

Figure 4: Effects of multimodal fusion.   Figure 5: Effects of the PSAR strategy.

**Hyperparameter study.** We conducted experiments to investigate the impact of hyperparameters. The detailed results, provided in Appendix H, indicate that both the number of multimodal fusion layers and the configuration of bottlenecks can influence the performance of multimodal fusion. For the problems we study, $L' = 3$, $n_b = n'_b = 10$, and $w = h = 16$ are the desirable settings.

## 6   CONCLUSION

This paper proposes a generic GIMF framework that integrates complementary graph and image information for neural MOCO. We first construct an instance image to introduce structured image data, and the PSAR strategy during image construction is designed to enhance out-of-distribution generalization. Then, we develop a multimodal fusion mechanism with MSB to efficiently fuse graph and image information. Our GIMF framework is deployed with two state-of-the-art neural MOCO methods. Experimental results confirm its effectiveness, and the ablation study highlights the necessity of both PSAR and MSB, particularly for improving out-of-distribution performance. A limitation of our current approach is its generalization performance on real-world scenarios and various MOCO problems. In future work, we plan to address this by exploring the hierarchical approach (Goh et al., 2024) and multi-task learning technique (Zhou et al., 2024).

Another potential limitation of our method is that constructing images requires problem-specific designs to a certain extent, and incorporating images slightly increases the demand for computational resources. However, the resulting performance improvements are substantial, making this trade-off worthwhile. Furthermore, while the problems studied in this paper are well-suited for image representation, there may be some other MOCO problems for which generating simple yet meaningful images is challenging. In these cases, the applicability of our method remains an open question.

ACKNOWLEDGMENTS AND DISCLOSURE OF FUNDING

This work is supported in part by the National Natural Science Foundation of China (62472461), and the Guangdong Basic and Applied Basic Research Foundation (2025A1515010129); in part by the National Research Foundation, Singapore under its AI Singapore Programme (AISG Award No. AISG3-RP-2022-031).

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

## A  DETAILED STATEMENTS OF THE STUDIED MOCO PROBLEMS

**Multi-objective traveling salesman problem (MOTSP)**   In the $M$-objective traveling salesman problem with $n$ nodes, each node $i \in 1, ..., n$ is associated with $M$ groups of 2-dimensional coordinates. The Euclidean distance $c_{ij}^m$ between nodes $i$ and $j$ is determined by their coordinates for objective $m$. The task is to find a tour $\boldsymbol{\pi}$ that visits all nodes while minimizing the total distances for each of the $M$ objectives. Specifically, the objective is to minimize $\boldsymbol{f}(\boldsymbol{\pi}) = (f_1(\boldsymbol{\pi}), f_2(\boldsymbol{\pi}), \ldots, f_M(\boldsymbol{\pi}))$, where $f_m(\boldsymbol{\pi}) = \sum_{i=1}^{n-1} c_{\pi_i, \pi_{i+1}}^m + c_{\pi_n, \pi_1}^m, \forall m \in \{1, \ldots, M\}$. The instances are generated by uniformly sampling coordinates within the range $[0, 1]^{2M}$.

**Multi-objective capacitated vehicle routing problem (MOCVRP)**   We study the bi-objective capacitated vehicle routing problem (Bi-CVRP), which involves $n$ customer nodes and one depot node. Each node has a 2-dimensional coordinate, and each customer is associated with a demand. A fleet of homogeneous vehicles, all with the same capacity, is based at the depot and must serve all customers before returning. When serving a customer, the vehicle's remaining capacity must be at least equal to the customer's demand. The two conflicting objectives are to minimize the total tour length and the makespan, defined as the longest route. For Bi-CVRP instances, the coordinates of both the depot and customers are uniformly sampled from the range $[0, 1]^2$, while demands are drawn from the set $\{1, \ldots, 9\}$. Vehicle capacity is set at 30, 40, and 50 for $20 \le n < 40$, $40 \le n < 70$, and $70 \le n \le 100$, respectively. All demands are normalized by the vehicle capacity for consistency.

**Multi-objective knapsack problem (MOKP)**   In the multi-objective knapsack problem with $M$ objectives and $n$ items, each item has a weight and $M$ distinct values. The items are represented as nodes in the instance graph. The objective is to select items that maximize all $M$ objectives simultaneously, while ensuring the total weight does not exceed the knapsack capacity. The instances are generated by sampling the weight and values of each item from a uniform distribution over $[0, 1]$. The knapsack capacity is set to 12.5 for $50 \le n < 100$ and 25 for $100 \le n \le 200$, respectively.

## B  ARCHITECTURE OF THE GRAPH TRANSFORMER

The MOCO instance graph can be processed by a graph Transformer (Kool et al., 2019) based on an encoder-decoder architecture. Given $n$ nodes with $z$-dimensional features $\boldsymbol{u}_1, \ldots, \boldsymbol{u}_n \in \mathcal{R}^z$, the encoder first transforms them to the initial node embeddings $\boldsymbol{h}_1^{(0)}, \ldots, \boldsymbol{h}_n^{(0)} \in \mathcal{R}^d$ via linear projection as $\boldsymbol{h}_i^{(0)} = W^u \boldsymbol{u}_i + \boldsymbol{b}^u, \forall i \in \{1, \ldots, n\}$, and then compute the final node embeddings $\boldsymbol{h}_1^{(L)}, \ldots, \boldsymbol{h}_n^{(L)}$ via $L$ Transformer layer. Each Transformer layer sequentially comprises a multi-head self-attention (MHSA) block with 8 attention heads, a residual connection and instance normalization (IN) block, a feed-forward (FF) block, and another residual connection and IN block. The node embeddings $\boldsymbol{H}^{(l)} = \{\boldsymbol{h}_1^{(l)}, \ldots, \boldsymbol{h}_n^{(l)}\}$ are updated as follows,

$$\hat{\boldsymbol{H}} = \text{IN}(\boldsymbol{H}^{(l-1)} + \text{MHSA}(\boldsymbol{H}^{(l-1)})) \tag{5}$$

$$\boldsymbol{H}^{(l)} = \text{IN}(\hat{\boldsymbol{H}} + \text{FF}(\hat{\boldsymbol{H}})). \tag{6}$$

The decoder take the derived node embeddings as inputs to calculate the selection probabilities for candidate nodes in an autoregressive manner with $T$ steps. Specifically, at decoding step $t \in \{1, ..., T\}$, the *glimpse* $\boldsymbol{q}_c$ is first computed using a problem-specific *context* embedding $\boldsymbol{h}_c$, as follows,

$$\boldsymbol{q}_c = \text{MHCA}(\boldsymbol{h}_c, \boldsymbol{H}^{(L)}), \tag{7}$$

where $\text{MHCA}(\boldsymbol{X}, \boldsymbol{Y})$ the multi-head cross-attention with $\boldsymbol{X}$ as the *querys* and with $\boldsymbol{Y}$ as the *keys* and *values*. The number of attention heads is set to 8. The definition of $\boldsymbol{h}_c$ is provided in Appendix C. Then, the *compatibility* is calculated by the attention mechanism, as follows,

$$\alpha_i = \begin{cases} -\infty, & \text{if node } i \text{ is masked} \\ C \cdot \tanh(\frac{\boldsymbol{q}_c^T (W^K \boldsymbol{h}_i^{(L)})}{\sqrt{d}}), & \text{otherwise} \end{cases} \tag{8}$$

where $C = 10$ is use to clip the result. Finally, the probabilities of node selection is obtained using a softmax function.

## C CONTEXT EMBEDDING

For MOTSP, the context embedding $h_c$ at each decoding step is constructed by concatenating the embeddings of the first and last visited nodes, with all visited nodes masked when calculating node selection probabilities. In MOCVRP, the context embedding $h_c$ consists of the embedding of the last visited node and the remaining vehicle capacity, and nodes already visited or with demands exceeding the remaining capacity are masked during probability computation. For MOKP, the context embedding $h_c$ combines the graph embedding $\bar{h} = \sum_{i=1}^{n} h_i / n$ with the remaining knapsack capacity, masking selected items and those with weights larger than the remaining capacity during probability calculations.

## D ARCHITECTURES OF GIMF-C AND GIMF-P

### D.1 GIMF-C

Following CNH (Fan et al., 2024), GIMF-C also employs a multi-head dual-attention (MHDA) mechanism to handle the given weight vector and uses a size-aware decoder to capture the feature of the problem size. Thus, we adopt the multi-head dual-attention mechanism with 8 attention heads for each modality in our multimodal fusion model. Specifically, the initial embedding of the weight vector $h_{n+1}^{(0)}$, node embeddings $\boldsymbol{H} = \{h_1^{(0)}, \ldots, h_n^{(0)}\}$, and patch embeddings $\boldsymbol{H}' = \{h_1'^{(0)}, \ldots, h_n'^{(0)}\}$ are first obtained using separate linear projections. Then, in each single-modal Transformer layer $l, \forall l \in \{1, \ldots, L - L'\}$, the multi-head self-attention (MHSA) is replaced by the MHDA, as follows,

$$\{\hat{\boldsymbol{H}}, \hat{h}_{n+1}\} = \text{IN}(\{\boldsymbol{H}^{(l-1)}, h_{n+1}^{(l-1)}\} + \text{MHDA}(\boldsymbol{H}^{(l-1)}, h_{n+1}^{(l-1)})), \tag{9}$$

$$\{\boldsymbol{H}^{(l)}, h_{n+1}^{(l)}\} = \text{IN}(\{\hat{\boldsymbol{H}}, \hat{h}_{n+1}\} + \text{FF}(\{\hat{\boldsymbol{H}}, \hat{h}_{n+1}\})), \tag{10}$$

$$\{\hat{\boldsymbol{H}}', \hat{h}'_{n+1}\} = \text{IN}(\{\boldsymbol{H}'^{(l-1)}, h_{n+1}'^{(l-1)}\} + \text{MHDA}(\boldsymbol{H}'^{(l-1)}, h_{n+1}'^{(l-1)})), \tag{11}$$

$$\{\boldsymbol{H}'^{(l)}, h_{n+1}'^{(l)}\} = \text{IN}(\{\hat{\boldsymbol{H}}', \hat{h}'_{n+1}\} + \text{FF}(\{\hat{\boldsymbol{H}}', \hat{h}'_{n+1}\})), \tag{12}$$

where $h_{n+1}'^{(0)} = h_{n+1}^{(0)}$. MHDA$(\boldsymbol{X}, \boldsymbol{Y})$ is formulated by integrating the MHSA and multi-head cross-attention (MHCA) mechanisms, as follows,

$$\text{MHDA}(\boldsymbol{X}, \boldsymbol{Y}) = \{(\text{MHSA}(\{\boldsymbol{X}, \boldsymbol{Y}\})_{:-1} + \text{MHCA}(\boldsymbol{X}, \boldsymbol{Y})), \text{MHSA}(\{\boldsymbol{X}, \boldsymbol{Y}\})_{-1}\}. \tag{13}$$

After $L - L'$ single-modal Transformer layers, the embeddings are updated by $L'$ multimodal fusion layers. The multimodal fusion layers involve additional $n_b$ graph bottlenecks $\boldsymbol{B} = \{b_1, \ldots, b_{n_b}\}$ and $n_b'$ image bottlenecks $\boldsymbol{B}' = \{b_1', \ldots, b_{n_b'}'\}$. In each layer $l, \forall l \in \{L - L' + 1, \ldots, L\}$, the embeddings are updated through the graph-guided and image-guided dual-attention. In this process, the MHSA mechanism in MHDA (see Equation 13) is substituted with our multimodal multi-head cross-attention mechanism (see Equation 3), as follows,

$$\tilde{\boldsymbol{H}} = \text{MHCA}(\{\boldsymbol{B}^{(l-1)}, \boldsymbol{H}^{(l-1)}, h_{n+1}^{(l-1)}\}, \{\boldsymbol{B}^{(l-1)}, \boldsymbol{H}^{(l-1)}, h_{n+1}^{(l-1)}, \boldsymbol{B}'^{(l-1)}\}) \tag{14}$$

$$\{\boldsymbol{B}^{(l)}, \boldsymbol{H}^{(l)}, h_{n+1}^{(l)}\} = \{\tilde{\boldsymbol{H}}_{:n_b}, (\tilde{\boldsymbol{H}}_{n_b:-1} + \text{MHCA}(\boldsymbol{H}^{(l-1)}, h_{n+1}^{(l-1)})), \tilde{\boldsymbol{H}}_{-1}\}, \tag{15}$$

$$\tilde{\boldsymbol{H}}' = \text{MHCA}(\{\boldsymbol{B}'^{(l-1)}, \boldsymbol{H}'^{(l-1)}, h_{n+1}'^{(l-1)}\}, \{\boldsymbol{B}'^{(l-1)}, \boldsymbol{H}'^{(l-1)}, h_{n+1}'^{(l-1)}, \boldsymbol{B}^{(l-1)}\}) \tag{16}$$

$$\{\boldsymbol{B}'^{(l)}, \boldsymbol{H}'^{(l)}, h_{n+1}'^{(l)}\} = \{\tilde{\boldsymbol{H}}'_{:n_b'}, (\tilde{\boldsymbol{H}}'_{n_b':-1} + \text{MHCA}(\boldsymbol{H}'^{(l-1)}, h_{n+1}'^{(l-1)})), \tilde{\boldsymbol{H}}'_{-1}\}. \tag{17}$$

In the decoder, the size-injected node embeddings $\bar{\boldsymbol{H}}^{(L)}$ are obtained by adding the problem size embedding (Fan et al., 2024) to the node embeddings. Then, the glimpse is derived as follows,

$$q_c = \text{MHCA}(h_c, \{\bar{\boldsymbol{H}}^{(L)}, \boldsymbol{H}'^{(L)}\}). \tag{18}$$

Finally, the probabilities of node selection can be computed as the same.

Table 4: Reference points and ideal points for the MOCO problems.

| Problem | Size | $r$ | $z$ |
|---|---|---|---|
| Bi-TSP | 20 | (20, 20) | (0, 0) |
| | 50 | (35, 35) | (0, 0) |
| | 100 | (65, 65) | (0, 0) |
| | 150 | (85, 85) | (0, 0) |
| | 200 | (115, 115) | (0, 0) |
| Bi-CVRP | 20 | (30, 4) | (0, 0) |
| | 50 | (45, 4) | (0, 0) |
| | 100 | (80, 4) | (0, 0) |
| Bi-KP | 50 | (5, 5) | (30, 30) |
| | 100 | (20, 20) | (50, 50) |
| | 200 | (30, 30) | (75, 75) |
| Tri-TSP | 20 | (20, 20, 20) | (0, 0) |
| | 50 | (35, 35, 35) | (0, 0) |
| | 100 | (65, 65, 65) | (0, 0) |

## D.2 GIMF-P

Following PMOCO (Lin et al., 2022), GIMF-P directly uses the multimodal fusion model as the base model. To manage each scalarized suproblem associated with a weight vector, GIMF-P employs a hypernetwork with a multi-layer perceptron (MLP) structure to generate decoder parameters, using the weight vector as input.

## E HYPERVOLUME

Hypervolume (HV) is a widely used indicator for evaluating the performance of MOCO methods, as it effectively measures both the convergence and diversity of the obtained Pareto front without requiring ground truth. The HV of a Pareto front $\mathcal{F}$ with respect to a reference point $r \in \mathcal{R}^M$, denoted as $\text{HV}_r(\mathcal{F})$, is defined as:

$$\text{HV}_r(\mathcal{F}) = \mu \left( \bigcup_{f(\pi) \in \mathcal{F}} [f(\pi), r] \right), \tag{19}$$

where $\mu$ represents the Lebesgue measure, and $[f(\pi), r]$ refers to an $M$-dimensional hypercube, i.e., $[f(\pi), r] = [f_1(\pi), r_1] \times \cdots \times [f_M(\pi), r_M]$. The HV is normalized as $\text{HV}'r(\mathcal{F}) = \text{HV}r(\mathcal{F})/\prod_{i=1}^M |r_i - z_i|$, where $z$ is an ideal point such that $z_i < \min\{f_i(\pi)|f(\pi) \in \mathcal{F}\}$ (or $z_i > \max\{f_i(\pi)|f(\pi) \in \mathcal{F}\}$ for maximization), $\forall i \in \{1, \ldots, M\}$. According to prior literature (Chen et al., 2023a;b), the same $r$ and $z$ are used across all methods for a given MOCO problem, as summarized in Table 4.

## F MULTI-OBJECTIVE INSTANCE AUGMENTATION

During the inference phase, instance augmentation (Lin et al., 2022) can be used to improve performance by transforming an instance into multiple variations that all share the same optimal solution. Each transformed instance is then solved, and the best solution among them is chosen. For Bi-CVRP, there are 8 possible transformations based on the 2-dimensional coordinates, such as $(x, y), (y, x), (x, 1 - y), (y, 1 - x), (1 - x, y), (1 - y, x), (1 - x, 1 - y), (1 - y, 1 - x)$. For the $M$-objective TSP, this leads to $8^M$ transformations, as each of the $M$ groups of coordinates can be permuted independently. For KP, this instance augmentation is not applicable.

Table 5: Detailed results on benchmark instances.

| Method | KroAB100 | | | KroAB150 | | | KroAB200 | | |
|---|---|---|---|---|---|---|---|---|---|
| | HV↑ | Gap↓ | Time↓ | HV↑ | Gap↓ | Time↓ | HV↑ | Gap↓ | Time↓ |
| WS-LKH | **0.7022** | **-0.30%** | 2.3m | **0.7017** | **-1.01%** | 4.0m | **0.7430** | **-1.43%** | 5.6m |
| MOEA/D | 0.6836 | 2.36% | 5.8m | 0.6710 | 3.41% | 7.1m | 0.7106 | 2.99% | 7.3m |
| NSGA-II | 0.6676 | 4.64% | 7.0m | 0.6552 | 5.69% | 7.9m | 0.7011 | 4.29% | 8.4m |
| MOGLS | 0.6817 | 2.63% | 52m | 0.6671 | 3.97% | 1.3h | 0.7083 | 3.30% | 1.6h |
| PPLS/D-C | 0.6785 | 3.09% | 38m | 0.6659 | 4.15% | 1.4h | 0.7100 | 3.07% | 3.8h |
| DRL-MOA | 0.6903 | 1.40% | 10s | 0.6794 | 2.20% | 18s | 0.7185 | 1.91% | 23s |
| MDRL | 0.6881 | 1.71% | 10s | 0.6831 | 1.67% | 17s | 0.7209 | 1.58% | 23s |
| EMNH | 0.6900 | 1.44% | 9s | 0.6832 | 1.66% | 16s | 0.7217 | 1.47% | 23s |
| PMOCO | 0.6878 | 1.76% | 9s | 0.6819 | 1.84% | 17s | 0.7193 | 1.80% | 23s |
| **GIMF-P** | 0.6936 | 0.93% | 11s | 0.6878 | 0.99% | 21s | 0.7231 | 1.28% | 26s |
| CNH | 0.6947 | 0.77% | 25s | 0.6892 | 0.79% | 35s | 0.7250 | 1.02% | 41s |
| **GIMF-C** | 0.6972 | 0.41% | 25s | 0.6895 | 0.75% | 35s | 0.7297 | 0.38% | 42s |
| MDRL-Aug | 0.6950 | 0.73% | 13s | 0.6890 | 0.82% | 19s | 0.7261 | 0.87% | 28s |
| EMNH-Aug | 0.6958 | 0.61% | 12s | 0.6892 | 0.79% | 18s | 0.7270 | 0.75% | 27s |
| PMOCO-Aug | 0.6937 | 0.91% | 12s | 0.6886 | 0.88% | 19s | 0.7251 | 1.01% | 32s |
| **GIMF-P-Aug** | 0.6971 | 0.43% | 15s | 0.6924 | 0.33% | 26s | 0.7271 | 0.74% | 45s |
| CNH-Aug | 0.6980 | 0.30% | 30s | 0.6938 | 0.13% | 37s | 0.7303 | 0.30% | 54s |
| **GIMF-C-Aug** | 0.7001 | 0.00% | 30s | 0.6947 | 0.00% | 39s | 0.7325 | 0.00% | 59s |

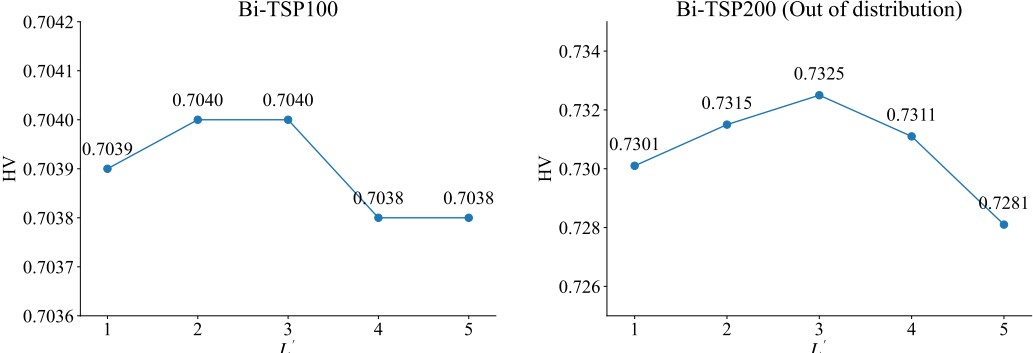

Figure 6: Effects of the number of fusion layers.

## G GENERALIZATION RESULTS ON BENCHMARK INSTANCES

The detailed out-of-distribution generalization results are presented in Table 5, further confirming the exceptional generalization ability of our GIMF.

## H HYPERPARAMETER STUDY

**Effects of the number of multimodal fusion layers** We vary the number of multimodal fusion layers $L'$ while keeping the total number of layers fixed at $L = 6$. The results in Figure 6 show that the model performs best when $L' = 3$. If $L'$ is too small, there may be insufficient fusion of multimodal information. Conversely, if $L'$ is too large, the model's ability to learn from single modalities may weaken, reducing its performance. Therefore, selecting an appropriate value for $L'$ is essential to achieve a balance between multimodal fusion and single-modality learning, ultimately improving the model's performance.

**Effects of the number of bottlenecks** For the number $n_b$ (where we set $n_b' = n_b$) of bottlenecks introduced for multimodal fusion, the model performs best when $n_b$ is set to 10, as shown in Figure 7. Introducing too many bottlenecks can increase the model's complexity, making effective learning

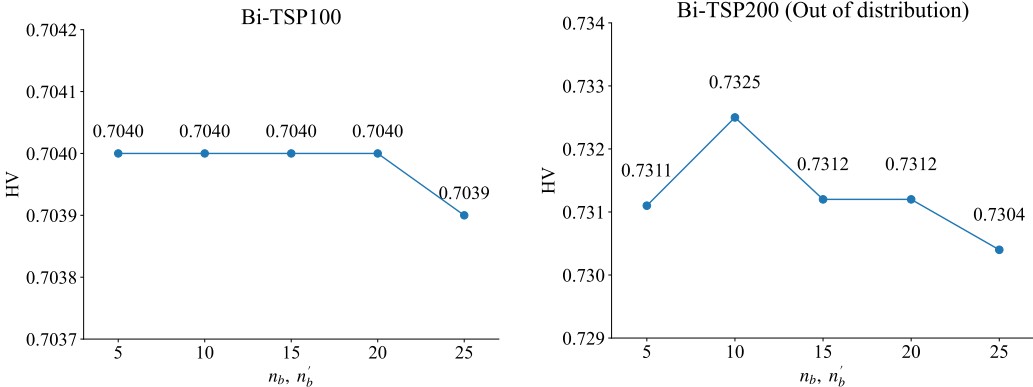

Figure 7: Effects of the number of bottlenecks.

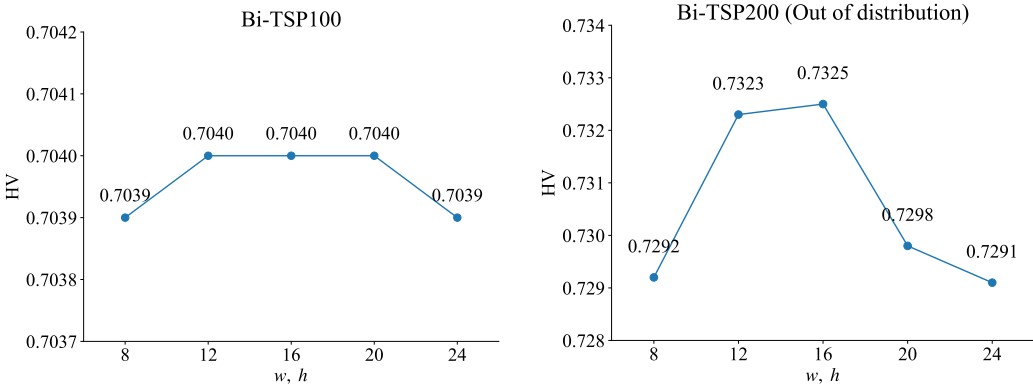

Figure 8: Effects of the patch size.

more difficult, while too few bottlenecks may hinder the proper fusion of multimodal information, limiting the model's capabilities. Thus, selecting an appropriate number of bottlenecks ensures effective multimodal fusion and improves the model's performance.

**Effects of the patch size** Different patch sizes correspond to different ways of dividing the image, and selecting an appropriate patch size is crucial. A large patch size results in a smaller number of coarse-grained patches, which may overlook some global information from the image, thereby affecting performance negatively. Conversely, a small patch size provides a finer-grained division, capturing more local details, but the resulting increase in the number of patches can raise learning difficulty and computational cost, ultimately harming performance. As shown in Figure 8, both excessively large and excessively small patch sizes lead to performance deterioration. Setting $w = h = 16$ strikes a desirable balance and yields the best results.

## I ANALYSIS OF NEURAL NETWORK SIZE

While incorporating additional images increases neural network size, the experimental results demonstrate considerable performance improvements that justify this trade-off. To further reinforce this conclusion, comparisons between GIMF-C (or GIMF-P) and CNH (or PMOCO) under comparable model parameter settings are presented in Table 6. Specifically, we control a similar number of model parameters by adjusting the embedding dimension $d$, which was originally set to 128 following the previous works such as CNH and PMOCO. The results strongly indicate that

Table 6: Comparisons between GIMF-C (or GIMF-P) and CNH (or PMOCO) under comparable model parameter settings.

| Method | | Bi-TSP20 | | Bi-TSP50 | | Bi-TSP100 | |
|---|---|---|---|---|---|---|---|
| Small-size | #(Parameter) | HV↑ | Gap↓ | HV↑ | Gap↓ | HV↑ | Gap↓ |
| PMOCO | 1.42M | 0.6259 | 0.18% | 0.6351 | 0.94% | 0.6957 | 1.53% |
| **GIMF-P-S** | 1.49M | 0.6265 | 0.08% | 0.6368 | 0.67% | 0.6993 | 1.02% |
| CNH | 1.63M | 0.6270 | 0.00% | 0.6387 | 0.37% | 0.7019 | 0.65% |
| **GIMF-C-S** | 1.66M | 0.6270 | 0.00% | 0.6391 | 0.31% | 0.7028 | 0.52% |
| Large-size | #(Parameter) | HV↑ | Gap↓ | HV↑ | Gap↓ | HV↑ | Gap↓ |
| PMOCO-L | 2.67M | 0.6265 | 0.08% | 0.6364 | 0.73% | 0.6988 | 1.09% |
| **GIMF-P** | 2.69M | 0.6266 | 0.06% | 0.6374 | 0.58% | 0.7006 | 0.84% |
| CNH-L | 3.22M | **0.6271** | **-0.02%** | 0.6393 | 0.28% | 0.7027 | 0.54% |
| **GIMF-C** | 3.20M | 0.6270 | 0.00% | **0.6397** | **0.22%** | **0.7040** | **0.35%** |

the performance improvements stem primarily from the GIMF framework itself, rather than merely from an increase in neural network size. The results are summarized as follows:

- We reduce $d$ to 80 for the small-size versions of GIMF-C and GIMF-P, referred to as GIMF-C-S and GIMF-P-S, aligning the number of their model parameters closely with CNH and PMOCO, respectively. The results, detailed in the table below, indicate that both GIMF-C-S and GIMF-P-S consistently outperform their respective backbones with the close number of model parameters.

- Besides, we explore a large-size case by raise $d$ to 200 for CNH and PMOCO, referred to as CNH-L and PMOCO-L, making the number of their model parameters close to GIMF-C and GIMF-P, respectively. The results on this large-size case consistently highlight the superiority of our GIMF.

- Notably, the small-size GIMF-P-S even surpasses the large-size PMOCO, while the small-size GIMF-C-S performs comparably to the large-size CNH.

