# OpenReview forum: "Neural Multi-Objective Combinatorial Optimization via Graph-Image Multimodal Fusion"
_ICLR.cc/2025/Conference — ICLR 2025 Poster_

### Official Review · Reviewer_4hxj · 2024-10-31

**Soundness:** 3
**Presentation:** 3
**Contribution:** 3
**Rating:** 6
**Confidence:** 3

**Summary:**

This paper introduces a novel approach for MOCO through graph-image multimodal fusion. The framework incorporates a constructed coordinate image and efficient multimodal fusion. Experimental results on MOCO problems show the advance of the proposed GIMF.

**Strengths:**

S1. The GIMF framework successfully combines graph and image information, enriching representation learning for MOCO problems.

S2. The PSAR strategy and modality-specific bottlenecks for multimodal fusion are well-justified and empirically validated.

S3. Extensive experiments are conducted.

**Weaknesses:**

W1. The improvement achieved by the proposed methods appears to be marginal.

W2. The computational cost of the proposed GIMF framework seems considerably higher than state-of-the-art methods like EMNH. It seems that the performance gains come from a cost of efficiency.

W3. Constructing images seems relatively straightforward for TSP problems, but how does this approach generalize to other real-world scenarios? For some tasks, image construction may be challenging—how do the authors envision addressing this limitation?

**Questions:**

N/A.

---

### Official Review · Reviewer_hYPJ · 2024-11-01

**Soundness:** 3
**Presentation:** 4
**Contribution:** 3
**Rating:** 8
**Confidence:** 3

**Summary:**

This paper proposes a generic graph-image multimodal fusion (GIMF) framework that integrates graph and image information of the problem instances to enhance neural MOCO. The framework consists of three main components: (1) a constructed coordinate image (2) a problem-size adaptive resolution strategy and (3) a multimodal fusion mechanism. Experimental results demonstrate its effectiveness.

**Strengths:**

1.	This paper is well-organized and clear writing.
2.	The proposed method demonstrates novelty.
3.	Experiments show that GIMF performs better on classic MOCO problems.

**Weaknesses:**

Some details and parameter settings were not explained clearly (see Questions below).

**Questions:**

1. What does "$\bm{\pi_i}$" represent in the formula for calculating $\nabla\mathcal{L}(\bm{\theta})$ on line 136, or should it be changed to "$\pi_i$"?

2. What is the meaning of the line from $\pi_t$ to $h_c$ in Figure 2? The authors should supplement the relationship between $\pi_t$ and $h_c$ in the main text.

3. Why choose the dimension of patch as $w=h=16$? The authors should provide an explanation or conduct ablation experiments.

---

### Official Review · Reviewer_SVeH · 2024-11-03

**Soundness:** 3
**Presentation:** 3
**Contribution:** 3
**Rating:** 6
**Confidence:** 3

**Summary:**

This paper presents a novel Graph-Image Multimodal Fusion (GIMF) framework designed to enhance multi-objective combinatorial optimization (MOCO) methods. By integrating both graph and image information from problem instances, the framework effectively addresses the limitations associated with relying solely on graph-modal information, particularly in the context of bi- and tri-objective traveling salesman problems (TSP).

**Strengths:**

- The main novelty of this paper lies in defining the image modality alongside the graph modality, offering a new perspective for addressing challenges in MOCO.
- This approach enhances conventional heuristic algorithms by leveraging the combined information from both modalities.

**Weaknesses:**

- If the image modality has been defined, why still continue to use the graph modality? How could one approach solving an MOCO problem using only the image modality? Additionally, could you provide ablation studies to support this?
- Since the graph can also be viewed as a transition matrix, what is the relationship between reinforcement learning and the authors' algorithm?

**Questions:**

- Is this analogous to a game played on a chessboard? Are the authors providing definitions and citations related to reinforcement learning? Are you aiming to transform the graph problem into a gameplay problem on a chessboard?
- Has reinforcement learning for MOCO in chessboard games been well studied elsewhere? If MOCO, such as TSP, is defined within the context of a chessboard game, isn’t it relatively straightforward? Does this require a graph modality, or can the image modality defined by the authors alone be sufficient to address the MOCO problems?

---

### Official Review · Reviewer_3Wbv · 2024-11-03

**Soundness:** 3
**Presentation:** 4
**Contribution:** 3
**Rating:** 8
**Confidence:** 3

**Summary:**

The paper presents a novel graph-image multimodal fusion (GIMF) framework that aims to enhance neural multi-objective combinatorial optimization (MOCO) methods. The GIMF framework integrates graph and image information of problem instances, which is designed to overcome the limitations of existing neural MOCO methods that rely solely on graph-modal information. The main contribution of the proposed method is the coordinate image construction, which provides complementary information to the graph representation.To improve the model's generalization across different problem sizes, a Problem-size Adaptive Resolution (PSAR) strategy is proposed during the image construction process, which helps maintain a stable density for both the image and patches. A multimodal fusion mechanism with Modality-Specific Bottlenecks (MSB) is designed to efficiently couple graph and image information.

The GIMF framework is implemented with two state-of-the-art neural MOCO backbones, namely CNH and PMOCO. Experimental results on classic MOCO problems demonstrate that GIMF can improve neural MOCO methods by providing image-modal information and exhibits superior generalization capability.

**Strengths:**

1. The major contribution of this paper is its integration of both graph and image modalities to enhance the representation learning for MOCO problems. The construction of coordinate images and the use of PSAR strategy are innovative steps that address the limitations of relying solely on graph information. The proposed MSB in multimodal fusion mechanism is also a novel contribution.

2. The paper is well-organized and written in a clear and concise manner. The introduction effectively sets the stage by outlining the challenges in MOCO and the motivation behind the GIMF framework. The preliminary section clearly describes the definition of the MOCO problem and related concepts, as well as the graph transformer for MOCO. The methodology section is detailed, providing a clear explanation of the image construction process, the PSAR strategy, and the multimodal fusion mechanism.

3. The significance mainly comes from its novelty.Specifically, leveraging a multimodal approach that incorporates image-modal information, which has the potential to improve many existing neural MOCO methods. The paper is also likely to inspire further research in constructing and learning from images of MOCO problems.

4. The experimental results suggest that GIMF does not obviously increase computational time of the neural MOCO basebone. The major innovations PSAR and MSB are validated by ablation study.

**Weaknesses:**

The proposed method could improve the performance of CNH and PMOCO, as well as their augment variants. However, sometimes the improvement seems marginal. In Table 1 and Table 2, the reported improvements are all mostly less than 0.001, and sometimes are as small as 0.0001. Meanwhile, the reported best results can not significantly outperform SOTA baselines.

**Questions:**

In Table 1 and Table 2, the reported times of GIMF-P and GIMF-C are sometimes smaller than PMOCO and CHN, what is the reason for this phenomenon?

---

### Official Review · Reviewer_QYyp · 2024-11-04

**Soundness:** 2
**Presentation:** 3
**Contribution:** 2
**Rating:** 5
**Confidence:** 2

**Summary:**

This paper aims to fully leverage the intrinsic features of problem instances by proposing a novel graph-image multimodal fusion framework for solving multi-objective combinatorial optimization (MOCO). The authors introduce the concept of "image" for MOCO to better capture the spatial structure of problem instances, enhancing the learning process. They also propose a problem-size adaptive resolution strategy to improve generalization. Finally, the paper presents a multimodal fusion mechanism with modality-specific bottlenecks to efficiently integrate graph and image information.

**Strengths:**

1. The writing is good.

2. The experiments are detailed, and the results are competitive.

**Weaknesses:**

1. I believe the role of the image concept in CO is questionable. To some extent, using images in CO results in information loss and requires more space to represent. As far as I know, many Euclidean TSP models, like [1, 2], use positions directly as input, which requires less space and provides more precise information.

2. Compared to typical neural MOCO methods, GIMF uses sparse matrix images as input, resulting in larger neural network sizes and an inability to handle larger-scale routing problems.

**Questions:**

1. What's the training loss of the GIMF?

2. Can GIMF obtain a Pareto set of solutions?

3. I noticed the authors used multi-modal fusion, but I don't quite understand this part. Does it mean that each subproblem requires training a single model and then performing model fusion at the end?

---

### Meta-Review · Area_Chair_fNAY · 2024-12-16

**Metareview:**

The paper proposes a graph-image multimodal fusion framework to enhance neural multi-objective combinatorial optimization.
Contributions: constructing coordinate images, employing a problem-size adaptive resolution strategy, and introducing a multimodal fusion mechanism with modality-specific bottlenecks.
Strengths: the experiments demonstrate consistent improvements over state-of-the-art method; novelty in integrating image-based information and detailed experimental validation.
Weaknesses: marginal improvements reported in certain cases and the increased computational cost due to image processing;  constructing images for some real-world problems might be challenging.

**Additional Comments On Reviewer Discussion:**

The responses addressed most concerns, leading to my acceptance recommendation based on the framework’s novel approach and experimental support.

---

### Decision · Program_Chairs · 2025-01-22

Accept (Poster)